# Lifestyle elements and risk of metabolic syndrome in adults

**Edyta Suliga**[1], **Elzbieta Ciesla**[1], **Magdalena Lelonek**[1]*, **Agnieszka Piechowska**[2], **Stanislaw Gluszek**[2]

1 Institute of Health Sciences, Medical College, Jan Kochanowski University, Kielce, Poland, 2 Institute of Medical Sciences, Medical College, Jan Kochanowski University, Kielce, Poland

* magdalena.lelonek@ujk.edu.pl

## Abstract

### Background

The aim of the study was to investigate which elements of lifestyle are associated with metabolic health in adults, defined as the absence of components of metabolic syndrome (MetS) based on the International Diabetes Federation criteria.

### Methods

Data from 10,277 individuals aged 40–65 years constituted the material of this study. Univariate and multivariate analyses with backward stepwise selection were carried out to identify the factors associated with the absence of metabolic disorders.

### Results

No family history of cardiovascular disease increased the odds of being metabolically healthy 1.5-2-fold. Furthermore, the following factors were associated with higher odds of being metabolically healthy in men: abstinence from alcohol (healthy individuals vs. those with ≥3 [OR = 5.49 (2.23–13.52); p<0.001], ≥2 [OR = 4.52 (1.87–10.91); p = 0.001], and ≥1 components of MetS [OR = 3.04 (1.41–6.56); p = 0.005]), moderate alcohol use (healthy individuals vs. those with ≥3 [OR = 3.36 (1.54–7.32); p = 0.002], ≥2 [OR = 3.28 (1.52–7.10); p = 0.002], and ≥1 components of MetS [OR = 3.93 (1.64–9.42); p = 0.002]), moderate-to-vigorous physical activity (MVPA) >2 hours/day and sitting time of 3–6 hours/day. Drinking >2 cups of coffee per day (healthy vs. those with ≥3 [OR = 2.00 (1.47–2.71)], ≥2 [OR = 1.84 (1.38–2.45)], and ≥1 components of MetS [OR = 1.72 (1.30–2.28); all p<0.001]), limiting animal fats to <2 servings/day, MVPA >2 hours/day, and 7–8 hours of sleep per day were associated with higher odds of being metabolically healthy in women.

### Conclusions

A healthy lifestyle increased the odds of being metabolically healthy. Some lifestyle modifications may appear effective in prevention of not only MetS, but also single metabolic risk factors.

**Data Availability Statement:** All relevant data are within the paper and its Supporting Information files.

**Funding:** We thank the Maria Sklodowska-Curie Institute of Oncology in Warsaw (Poland) and the

Polish-Norwegian Foundation Research Fund for support. Project financed under the program the Minister of Education and Science called "Regional Initiative of Excellence" in the years 2019-2023, project no. 024/RID/2018/19, amount of financing 11 999 000,00 PLN. The founders had no role in study design, data collection and analysis, decision to publish, or preparation of the manuscript.

**Competing interests:** The authors have declared that no competing interests exist.

## Introduction

Metabolic disorders, such as impaired glucose tolerance, insulin resistance, hyperinsulinemia, dyslipidemia, hypertension, and abdominal obesity, increase the risk of type 2 diabetes, cardiovascular disease, some cancers, and numerous other diseases [1, 2]. Current literature emphasizes that there is no single, universally accepted definition of metabolic health [3]. It is usually defined as the absence of metabolic syndrome (MetS), i.e., any three out of five components, including abdominal obesity, elevated blood pressure, increased glucose and triglyceride (TG) levels, and decreased high-density lipoprotein (HDL) levels [4]. Therefore, several studies considered individuals with ≤2 factors [2] or ≤1 factor [3] as healthy, whereas only few studies defined healthy individuals as those with none of the above-mentioned risk factors. Yet, the chance of developing cardiovascular disease increases proportionally with the number of risk factors an individual has [5]. In a prospective study, adjusted hazard ratios for MetS among individuals with one or two risk factors were 3 to 16 times higher than in those without risk factors at baseline [6]. Hence, the results of previously mentioned studies suggest that every person who has even one cardiometabolic risk factor should be classified as metabolically unhealthy [7].

Although several variants of different genes are involved in the pathogenesis of MetS, all of its components are influenced by lifestyle, i.e., mainly an unhealthy diet [8, 9] and insufficient physical activity [10]. The aim of this study was to investigate which elements of lifestyle are associated with metabolic health (absence of MetS components) in adults aged 40–65 years.

## Materials and methods

### Study population

Data from the PONS (POlish-Norwegian Study) project, which was carried out from 2010 to 2011, constituted the material of this study. The project aimed to assess the health of residents living in the city of Kielce and the surrounding rural area in Świętokrzyskie Voivodeship, Poland. Informed consent was obtained from all individual participants included in the study. People aged from 45 to 64 years were the target group; however, slightly younger and older people also volunteered to participate in the study. The response rate was 12%. For the purpose of this study, data from participants aged 40–65 years (median of 55.0 years, SD = 5.35) were used. Detailed information regarding the project protocol and group selection was described in paper published previously [8].

Study participants comprised 13,172 volunteers. After removing incomplete data (803 participants) and excluding individuals who reported having current or past history of cancer or stroke (2092 participants), the final number of individuals included in the analyses was 10,277, including 6,757 women; 5.87% of the participants were diagnosed with diabetes, 37.35% with hypertension and 16.46% with dyslipidemia.

The study was approved by the Ethics Committee from the Cancer Centre and Institute of Oncology in Warsaw, No. 69/2009/1/2011 (data collection) and the Committee on Bioethics at the Faculty of Health Sciences, Jan Kochanowski University in Kielce, Poland, No. 45/2016 (data analysis).

### Anthropometric measurements, assessment of biomarkers, blood pressure and lifestyle data

Basic anthropometric measurements, blood biochemical tests, and blood pressure measurements were carried out. Also, socio-demographic data and information on family history of

chronic diseases (non-communicable) and participants' lifestyle were obtained. All information was gathered via face-to-face interviews using structured questionnaires.

To measure body weight, a body composition analyzer (Tanita S.C. 240MA, Tokyo, Japan) was used. Body height measurements were performed by means of the weight stadiometer. Weight and height measurements were used to calculate the body mass index (BMI, kg/m$^2$). Waist circumference was measured halfway between the lower border of the rib and the upper iliac crest using a non-elastic metric tape measure.

Fasting glucose, cholesterol, and triglyceride (TG) levels were determined from blood samples. Serum glucose level was measured with hexokinase. HDL-cholesterol levels were determined using the colorimetric non-precipitation method with polyethylene glycol modified enzymes. TG levels were assessed enzymatically with glycerol phosphate oxidase (determination of $H_2O_2$ with peroxidase).

Blood pressure was measured with the use of the blood pressure monitor Omron (model M3 Intellisense, Mannheim, Germany). The test was carried out at the artery in the right upper limb, when seated, and the average of two measurements was used in the analyses.

Information on smoking, consumption of alcohol, coffee and other food products, physical activity, sedentary behavior, and sleep duration was self-reported and collected during a face-to-face interview. Dietary data were collected using the Food Frequency Questionnaire (FFQ), which consisted of a list of 67 products. The questionnaire was constructed based on a previously designed and validated FFQ for the Polish arm of the PURE study. The questionnaire was highly accurate and repeatable compared to the reference method [11]. The respondents declared how frequently they consumed a standard portion of each product over the preceding year.

The intake of sweets and sugary drinks was determined based on the consumption of: sugar, chocolate bars, candies, cakes and cookies, sweetened fruit juices, and cola and other soft drink products. In the red and processed meat groups, the consumption of beef, pork, liver, franks, ham, bacon, and sausage was analyzed. Whole grains included: dark bread, cereals, oats, and groats. In the fruit and vegetables group, the following were included: seasonal fruits, apples, bananas, oranges, green leafy vegetables, broccoli, beetroots, raw cabbage, carrots, garlic, onions, and tomatoes. Corresponding numerical values, obtained by dividing the consumption frequency by the number of days in a month or a week, were assigned to each consumption frequency. Regarding the consumption frequency, quartile groups from the lowest (Q1) to the highest (Q4) were created for each variable. The animal fats group included butter and lard. Two categories of the consumption frequency were established: >1 time per day and ≤1 time per day. Information on consumption of beer, wine, fruit-based, and strong/high-percentage alcoholic drinks, such as vodka, in the previous year allowed us to estimate the average consumption of pure ethanol and the frequency of its use. Due to differences in alcohol consumption among the sexes, women were divided into non-drinkers (not drinking during the previous 12 months) and those drinking 0.1–15.0 g and >15.0 g ethanol/day, while men were divided into non-drinkers and those reporting drinking 0.1–30.0 g and >30.0 g ethanol/day.Three categories were identified for coffee drinking: non-drinking, drinking up to 2 servings/day, and drinking above 2 servings/day. One serving was defined as a 250 ml cup.

Based on the declarations on smoking, respondents were categorized into three groups: current smokers (smoking every day), former smokers (individuals reporting having smoked every day in the past for at least 6 months), and never-smokers.

Physical activity, walking, and sitting time were evaluated using the Polish version of the IPAQ Long Form, prepared in accordance with the IPAQ Scientific Committee's recommendations. The duration of moderate-to-vigorous physical activity (MVPA), walking, and sitting time were calculated. Four categories of MPVA duration (≤30 minutes, >30 minutes to ≤1

hour, >1 hour to ≤2 hours, >2 hours per day), three categories of walking time (≤30 minutes, >30 minutes to ≤1 hour and >1 hour per day), and three categories of sitting time (≤3 hours, >3 hours to 6 hours, and >6 hours per day) were established. Sleep duration (<7 hours, 7–8 hours, and ≥9 hours per night) was evaluated based on the answers to the question: "on average, how many hours do you sleep each night?".

### The definition of metabolic health

Participants without any components of MetS according to the IDF Taskforce on Epidemiology and Prevention guidelines [4] were considered as metabolically healthy. Due to the lower cut-off points defining abdominal obesity, this classification allows earlier diagnosis of MetS, i.e., with less excess weight. It also more accurately identifies individuals with MetS who have normal BMI.

### Family history of cardiovascular disease

Family history of cardiovascular disease was defined as having at least two family members with a diagnosis of the coronary heart disease, ischemic disease, myocardial infarction, hypertension or stroke. Family medical history reflects both the genetic morbidity and the behavioral risk factors present in a given family.

### Statistical methods

Distributions of all categorical variables were compared between metabolically healthy and unhealthy individuals using a non-parametric chi-squared test. Analyses were conducted in three models. Healthy individuals (0 MetS components) were compared to those with ≥3, ≥2, and ≥1 components of MetS, respectively. Each analysis was performed separately for sexes. Univariate analyses were performed, but the data were not shown. Multivariate analyses with backward stepwise selection were then conducted to identify factors associated with greater odds of being metabolically healthy (0 components of MetS). The analyzed variables and reference groups (ref.) were as follows: family history of cardiovascular disease (ref.: the presence of cardiovascular disease), smoking (ref.: smokers), alcohol consumption (ref.: >15 g–women; >30g –men), drinking coffee (ref.: not at all), consumption of sweets and sugary drinks (ref.: Q4), animal fats (ref.: more than once per day), red and processed meat (ref.: Q4), vegetables and fruit (ref.: Q1), whole grains (ref.: Q1), moderate-to-vigorous physical activity (MVPA) (ref.: ≤30 minutes/day), walking (ref.: ≤30 minutes/day), sitting time (ref.: >6 hours/day), and duration of sleep (ref.: <7 hours). Next, using backward stepwise selection, non-significant variables were eliminated until only those significant for maintaining metabolic health remained in the model. Statistical significance was defined as a p-value of <0.05. All statistical analyses were performed using STATISTICA (version 13.3, STATSOFT, PL).

## Results

The proportion of individuals with a family history of cardiovascular disease increased along with the number of MetS components (p<0.001). The lowest proportion of participants with a family history of cardiovascular disease was found in healthy men (8.60%) (Table 1), followed by women (16.57%) (Table 2). Among men, the highest proportion of never-smokers was found in healthy individuals (43.01%). Among healthy women, the proportion of former smokers was the lowest (22.99%), with the highest proportion of current smokers. Complete abstinence from alcohol was most frequently reported by healthy men (11.29%), while it was least frequently reported by those with ≥3 components of MetS (6.92%). In addition, the

**Table 1. Relationships between lifestyle, family history of cardiovascular disease, and prevalence of MetS components in men.**

| Factor | | Healthy individuals (0 MetS components) N | (%) | 1 MetS component N | (%) | 2 MetS components N | (%) | ≥3 MetS components N | (%) | p |
|---|---|---|---|---|---|---|---|---|---|---|
| family history of cardiovascular disease | yes | 16 | 8.60 | 83 | 14.04 | 134 | 13.56 | 284 | 16.66 | 0.008 |
| | no | 170 | 91.40 | 508 | 85.96 | 854 | 86.44 | 1421 | 83.34 | |
| smoking | current smoker | 48 | 25.81 | 162 | 27.41 | 202 | 20.45 | 346 | 20.29 | <0.001 |
| | former smoker | 58 | 31.18 | 193 | 32.66 | 382 | 38.66 | 808 | 47.39 | |
| | never-smoker | 80 | 43.01 | 236 | 39.93 | 404 | 40.89 | 551 | 32.32 | |
| consumption of alcohol | >30 g | 7 | 3.76 | 45 | 7.61 | 96 | 9.72 | 204 | 11.96 | <0.001 |
| | ≤30 g | 158 | 84.95 | 486 | 82.23 | 802 | 81.17 | 1383 | 81.11 | |
| | 0 g | 21 | 11.29 | 60 | 10.15 | 90 | 9.11 | 118 | 6.92 | |
| consumption of coffee | none | 32 | 17.20 | 117 | 19.80 | 195 | 19.74 | 392 | 22.99 | 0.109 |
| | ≤2 servings/ day | 90 | 48.39 | 284 | 48.05 | 485 | 49.09 | 835 | 48.97 | |
| | >2 servings/ day | 64 | 34.41 | 190 | 32.15 | 308 | 31.17 | 478 | 28.04 | |
| consumption of sweets and sugary drinks | Q1 | 34 | 18.28 | 117 | 19.80 | 228 | 23.08 | 484 | 28.39 | <0.001 |
| | Q2 | 48 | 25.81 | 163 | 27.58 | 245 | 24.80 | 407 | 23.87 | |
| | Q3 | 43 | 23.12 | 147 | 24.87 | 263 | 26.62 | 423 | 24.81 | |
| | Q4 | 61 | 32.80 | 164 | 27.75 | 252 | 25.51 | 391 | 22.93 | |
| consumption of animal fats | >1 time/day | 104 | 55.91 | 335 | 56.68 | 582 | 58.91 | 1073 | 62.93 | 0.015 |
| | ≤1 time/day | 82 | 44.09 | 256 | 43.32 | 406 | 41.09 | 632 | 37.07 | |
| consumption of red and processed meat | Q1 | 57 | 30.65 | 177 | 29.95 | 232 | 23.48 | 400 | 23.46 | 0.025 |
| | Q2 | 44 | 23.66 | 144 | 24.37 | 241 | 24.39 | 439 | 25.75 | |
| | Q3 | 42 | 22.58 | 139 | 23.52 | 272 | 27.53 | 414 | 24.28 | |
| | Q4 | 43 | 23.12 | 131 | 22.17 | 243 | 24.60 | 452 | 26.51 | |
| consumption of whole grains | Q1 | 53 | 28.49 | 154 | 26.06 | 230 | 23.28 | 360 | 21.11 | 0.051 |
| | Q2 | 45 | 24.19 | 151 | 25.55 | 278 | 28.14 | 421 | 24.69 | |
| | Q3 | 43 | 23.12 | 142 | 24.03 | 242 | 24.49 | 449 | 26.33 | |
| | Q4 | 45 | 24.19 | 144 | 24.37 | 238 | 24.09 | 475 | 27.86 | |
| consumption of fruit and vegetables | Q1 | 58 | 31.18 | 163 | 27.58 | 227 | 22.98 | 419 | 24.57 | 0.231 |
| | Q2 | 35 | 18.82 | 142 | 24.03 | 258 | 26.11 | 432 | 25.34 | |
| | Q3 | 52 | 27.96 | 141 | 23.86 | 249 | 25.20 | 425 | 24.93 | |
| | Q4 | 41 | 22.04 | 145 | 24.53 | 254 | 25.71 | 429 | 25.16 | |
| MVPA | ≤ 30 minutes/day | 59 | 31.72 | 173 | 29.27 | 318 | 32.19 | 664 | 38.94 | <0.001 |
| | >30 minutes to ≤ 1 hour/day | 30 | 16.13 | 122 | 20.64 | 162 | 16.40 | 306 | 17.95 | |
| | >1 hour to ≤ 2 hours/day | 32 | 17.20 | 118 | 19.97 | 191 | 19.33 | 305 | 17.89 | |
| | >2 hours/ day | 65 | 34.95 | 178 | 30.12 | 317 | 32.09 | 430 | 25.22 | |
| Walk | ≤ 30 minutes/day | 75 | 40.32 | 238 | 40.27 | 385 | 38.97 | 695 | 40.76 | 0.332 |
| | > 30 minutes to ≤1 hour/day | 30 | 16.13 | 111 | 18.78 | 193 | 19.53 | 363 | 21.29 | |
| | >1 hour/day | 81 | 43.55 | 242 | 40.95 | 410 | 41.50 | 647 | 37.95 | |
| sitting time | >6 hours/day | 44 | 23.66 | 191 | 32.32 | 316 | 31.98 | 579 | 33.96 | 0.185 |
| | >3 to ≤6 hours/day | 105 | 56.45 | 289 | 48.90 | 487 | 49.29 | 827 | 48.50 | |
| | ≤3 hours/day | 37 | 19.89 | 111 | 18.78 | 185 | 18.72 | 299 | 17.54 | |
| sleep duration | <7 hours/day | 54 | 29.03 | 153 | 25.89 | 233 | 23.58 | 422 | 24.75 | 0.152 |
| | 7–8 hours/day | 125 | 67.20 | 418 | 70.73 | 694 | 70.24 | 1185 | 69.50 | |
| | ≥9 hours/day | 7 | 3.76 | 20 | 3.38 | 61 | 6.17 | 98 | 5.75 | |

MetS–metabolic syndrome, p–value, Q–quintile, MVPA–moderate-to-vigorous physical activity.

**Table 2. Relationships between lifestyle, family history of cardiovascular disease, and prevalence of MetS components in women.**

| Factor | | Healthy individuals (0 MetS components) | | 1 MetS component | | 2 MetS components | | ≥3 MetS components | | p |
|---|---|---|---|---|---|---|---|---|---|---|
| | | N | (%) | N | (%) | N | (%) | N | (%) | |
| family history of cardiovascular disease | yes | 111 | 16.57 | 276 | 18.74 | 439 | 22.24 | 675 | 25.57 | <0.001 |
| | no | 559 | 83.43 | 1197 | 81.26 | 1535 | 77.76 | 1965 | 74.43 | |
| smoking | current smoker | 154 | 22.99 | 293 | 19.89 | 338 | 17.12 | 521 | 19.73 | 0.019 |
| | former smoker | 169 | 25.22 | 407 | 27.63 | 596 | 30.19 | 753 | 28.52 | |
| | never-smoker | 347 | 51.79 | 773 | 52.48 | 1040 | 52.68 | 1366 | 51.74 | |
| consumption of alcohol | >15 g | 18 | 2.69 | 38 | 2.58 | 46 | 2.33 | 44 | 1.67 | 0.005 |
| | ≤15 g | 552 | 82.39 | 1231 | 83.64 | 1601 | 81.10 | 2118 | 80.23 | |
| | 0 g | 100 | 14.93 | 203 | 13.79 | 327 | 16.57 | 478 | 18.11 | |
| consumption of coffee | none | 70 | 10.45 | 183 | 12.42 | 252 | 12.77 | 426 | 16.14 | <0.001 |
| | ≤2 servings/ day | 328 | 48.96 | 742 | 50.37 | 1080 | 54.71 | 1481 | 56.10 | |
| | >2 servings/ day | 272 | 40.60 | 548 | 37.20 | 642 | 32.52 | 733 | 27.77 | |
| consumption of sweets and sugary drinks | Q1 | 141 | 21.04 | 315 | 21.40 | 464 | 23.51 | 730 | 27.65 | <0.001 |
| | Q2 | 169 | 25.22 | 366 | 24.86 | 472 | 23.91 | 658 | 24.92 | |
| | Q3 | 175 | 26.12 | 367 | 24.93 | 533 | 27.00 | 664 | 25.15 | |
| | Q4 | 185 | 27.61 | 424 | 28.80 | 505 | 25.58 | 588 | 22.27 | |
| consumption of animal fats | >1 time/day | 393 | 58.66 | 851 | 57.77 | 1216 | 61.60 | 1718 | 65.08 | <0.001 |
| | ≤1 time/day | 277 | 41.34 | 622 | 42.23 | 758 | 38.40 | 922 | 34.92 | |
| consumption of red and processed meat | Q1 | 170 | 25.37 | 369 | 25.07 | 447 | 22.64 | 639 | 24.20 | 0.576 |
| | Q2 | 157 | 23.43 | 349 | 23.71 | 494 | 25.03 | 676 | 25.61 | |
| | Q3 | 169 | 25.22 | 384 | 26.09 | 542 | 27.47 | 662 | 25.08 | |
| | Q4 | 174 | 25.97 | 370 | 25.14 | 491 | 24.87 | 663 | 25.11 | |
| consumption of whole grains | Q1 | 164 | 24.48 | 380 | 25.82 | 466 | 23.61 | 627 | 23.75 | 0.483 |
| | Q2 | 157 | 23.43 | 353 | 23.98 | 511 | 25.89 | 707 | 26.78 | |
| | Q3 | 165 | 24.63 | 370 | 25.14 | 505 | 25.58 | 660 | 25.00 | |
| | Q4 | 184 | 27.46 | 369 | 25.07 | 492 | 24.92 | 646 | 24.47 | |
| consumption of fruit and vegetables | Q1 | 190 | 28.36 | 343 | 23.30 | 500 | 25.33 | 652 | 24.70 | 0.127 |
| | Q2 | 169 | 25.22 | 372 | 25.27 | 487 | 24.67 | 663 | 25.11 | |
| | Q3 | 141 | 21.04 | 393 | 26.70 | 513 | 25.99 | 642 | 24.32 | |
| | Q4 | 170 | 25.37 | 364 | 24.73 | 474 | 24.01 | 683 | 25.87 | |
| MVPA | ≤ 30 minutes/day | 148 | 22.09 | 339 | 23.01 | 470 | 23.81 | 706 | 26.74 | 0.042 |
| | >30 minutes to ≤ 1 hour/day | 179 | 26.72 | 372 | 25.25 | 516 | 26.14 | 684 | 25.91 | |
| | >1 hour to ≤ 2 hours/day | 151 | 22.54 | 374 | 25.39 | 488 | 24.72 | 623 | 23.60 | |
| | >2 hours/day | 192 | 28.66 | 388 | 26.34 | 500 | 25.33 | 627 | 23.75 | |
| walk | ≤ 30 minutes/day | 262 | 39.10 | 520 | 35.30 | 759 | 38.45 | 1101 | 41.70 | <0.001 |
| | >30 minutes to ≤1 hour/day | 140 | 20.90 | 337 | 22.88 | 450 | 22.80 | 621 | 23.52 | |
| | >1 hour/day | 268 | 40.00 | 616 | 41.82 | 765 | 38.75 | 918 | 34.77 | |
| sitting time | >6 hours/day | 212 | 31.64 | 473 | 32.11 | 567 | 28.72 | 712 | 26.97 | 0.020 |
| | >3 to ≤6 hours/day | 315 | 47.01 | 702 | 47.66 | 992 | 50.25 | 1352 | 51.21 | |
| | ≤3 hours/day | 143 | 21.34 | 298 | 20.23 | 415 | 21.02 | 576 | 21.82 | |
| sleep duration | <7 hours/day | 149 | 22.24 | 366 | 24.85 | 507 | 25.68 | 652 | 24.70 | 0.006 |
| | 7–8 hours/day | 486 | 72.54 | 1001 | 67.96 | 1314 | 66.57 | 1747 | 66.17 | |
| | ≥9 hours/day | 35 | 5.22 | 106 | 7.20 | 153 | 7.75 | 241 | 9.13 | |

MetS–metabolic syndrome, p—value, Q–quintile, MVPA—moderate-to-vigorous physical activity.

group with ≥3 components of MetS was characterized by the highest proportion of participants drinking >30 g alcohol per day (11.96%), whereas such a proportion was the lowest in healthy men (3.76%). Among women with ≥3 components of MetS, there were the most non-drinkers (18.11%) and the least of those who consumed >15 g alcohol/day.

Although no significant relationship between coffee consumption and the number of MetS components was found in men, the highest consumption of coffee was reported by men classified as healthy. In women, a significant negative correlation was found between coffee consumption and the number of MetS components. As many as 40.6% of healthy women consumed >2 servings of coffee/day. Men and women with ≥3 components of MetS consumed the least amounts of sweets and sugary drinks, while healthy participants reported the highest intake of these products. Animal fats as well as red and processed meat were most frequently consumed by men with the highest number of MetS components. A similar relationship, but only in relation to animal fat intake, was found in women. Relationships between the consumption of whole grains and fruits and vegetables and the number of MetS components in both sexes were not statistically significant.

The proportion of men reporting the longest time spent on MVPA was the highest among those classified as healthy (34.95%) and the lowest in the MetS group (25.22%). No significant differences were found between men in walking time, sitting time, and sleep duration depending on the number of MetS components. In women, there was a significant negative relationship between the time spent on MVPA, walking, and the number of MetS components. The proportion of healthy women who reported sitting time longer than 6 hours per day was relatively high (31.64%), whereas it was the lowest in those reporting sitting time of >3 to ≤6 hours per day (47.01%). Women classified as healthy most frequently reported sleep duration of 7–8 hours (72.54%), while women with MetS more often reported sleep duration of ≥9 hours (9.13%) compared to other female participants.

No family history of cardiovascular disease, abstinence from alcohol, and moderate alcohol use (≤30 g/day) vs. the overuse of alcohol (>30 g/day) as well as sitting time of 3 to 6 hours vs. sitting time of >6 hours/day (Tables 3–5) increased the odds of being metabolically healthy in men in each model (i.e., healthy individuals vs. those with ≥3, vs. ≥2 and vs. ≥1 components of MetS). Moreover, MVPA performed for >2 hours on a daily basis was a factor increasing the odds of being metabolically healthy compared to individuals with MetS (at least 3 out of 5 components).

Lower odds of being metabolically healthy in men were associated with: past smoking (compared to participants with ≥3 and ≥2 components of MetS), low consumption of sweets and sugary drinks (Q1 and Q3 vs. Q4; compared to participants with ≥3 and ≥1 components of MetS), and high consumption of fruit and vegetables (Q2 and Q4 vs. Q1; compared to participants with ≥3 and ≥2 components of MetS as well as Q2 vs. Q1; compared to participants with ≥1 component of MetS).

Factors significantly increasing the odds of being metabolically healthy in women in each model (healthy participants vs. those with ≥3, ≥2, and ≥1 components of MetS) included: no family history of cardiovascular disease, drinking >2 cups of coffee per day, and MVPA performed for >2 hours per day (Tables 3–5). Women limiting animal fats to ≤1 serving/day were more likely to be metabolically healthy compared to women with ≥3 and ≥2 components of MetS. Furthermore, women sleeping for 7–8 hours per day had higher odds of being metabolically healthy compared to women with ≥2 and ≥1 components of MetS. This tendency was also found in participants who walked for more than 1 hour per day, but only when compared to women with MetS (≥3 components).

Factors significantly lowering the odds of being metabolically healthy (when compared to women with MetS) were: sitting time of >3 to ≤6 hours/day, sleep duration of ≥9 hours/day,

**Table 3. Factors associated with metabolic health.** Multivariate analyses with backward stepwise selection results for ≥3 components of MetS in men and women.

| Lifestyle factor | | Men | | Women | |
|---|---|---|---|---|---|
| | | OR (95%CI) | p | OR (95%CI) | p |
| family history of cardiovascular disease | yes | **1.0** | - | **1.0** | - |
| | no | **2.02 (1.18–3.45)** | **0.011** | **1.71 (1.36–2.16)** | **<0.001** |
| smoking | current smoker | 1.0 | - | 1.0 | - |
| | former smoker | 0.58 (0.38–0.88) | 0.010 | - | - |
| | never-smoker | 1.15 (0.77–1.72) | 0.476 | - | - |
| consumption of alcohol | >30 g | 1.0 | - | 1.0 | - |
| | ≤30 g | **3.36 (1.54–7.32)** | **0.002** | 0.68 (0.38–1.24) | 0.213 |
| | 0 g | **5.49 (2.23–13.52)** | **<0.001** | 0.63 (0.33–1.18) | 0.149 |
| consumption of coffee | none | 1.0 | - | 1.0 | - |
| | ≤2 servings/ day | 1.35 (0.87–2.09) | 0.177 | 1.26 (0.94–1.68) | 0.127 |
| | >2 servings/ day | 1.57 (0.98–2.53) | 0.060 | **2.00 (1.47–2.71)** | **<0.001** |
| consumption of sweets and sugary drinks | Q1 | **0.48 (0.30–0.76)** | **0.002** | **0.70 (0.54–0.91)** | **0.007** |
| | Q2 | 0.76 (0.50–1.16) | 0.205 | 0.93 (0.72–1.20) | 0.575 |
| | Q3 | **0.63 (0.41–0.97)** | **0.036** | 0.91 (0.71–1.16) | 0.441 |
| | Q4 | 1.0 | - | 1.0 | - |
| consumption of whole grains | Q1 | 1.0 | - | 1.0 | - |
| | Q2 | 0.72 (0.47–1.11) | 0.133 | - | - |
| | Q3 | 0.74 (0.48–1.16) | 0.189 | - | - |
| | Q4 | 0.88 (0.56–1.38) | 0.578 | - | - |
| consumption of fruit and vegetables | Q1 | 1.0 | - | 1.0 | - |
| | Q2 | **0.56 (0.36–0.88)** | **0.012** | - | - |
| | Q3 | 0.81 (0.53–1.22) | 0.303 | - | - |
| | Q4 | **0.64 (0.41–0.99)** | **0.049** | - | - |
| MVPA | ≤ 30 minutes/day | 1.0 | - | 1.0 | - |
| | >30 minutes to ≤ 1 hour/day | 1.18 (0.74–1.90) | 0.475 | 1.13 (0.88–1.46) | 0.348 |
| | >1 hour to ≤ 2 hours/ day | 1.17 (0.73–1.87) | 0.500 | 1.04 (0.80–1.36) | 0.765 |
| | >2 hours/ day | **1.51 (1.02–2.24)** | **0.041** | 1.28 (0.99–1.66) | 0.062 |

(*Continued*)

**Table 3.** (Continued)

| Lifestyle factor | | Men | | Women | |
|---|---|---|---|---|---|
| | | OR (95%CI) | p | OR (95%CI) | p |
| walk | ≤ 30 minutes/day | 1.0 | - | 1.0 | - |
| | > 30 minutes to ≤1 hour/day | - | - | 1.01 (0.79–1.28) | 0.951 |
| | >1 hour/day | - | - | **1.23 (1.01–1.52)** | **0.045** |
| sleep duration | <7 hours/day | 1.0 | - | 1.0 | - |
| | 7–8 hours/day | - | - | 1.20 (0.97–1.49) | 0.090 |
| | <7 hours/day | - | - | **0.58 (0.37–0.89)** | **0.012** |

OR–odds ratio, CI–confidence interval, p—value, Q–quintile, MVPA—moderate-to-vigorous physical activity.

and low consumption of sweets and sugary drinks (Q1 vs. Q4). Lower odds of being metabolically healthy were also found in former smokers and in the case of vegetables and fruit consumption (Q1 vs. Q3), compared to women with ≥2 and ≥1 components of MetS.

## Discussion

The results of our study show that a healthy lifestyle may significantly increase the odds of being metabolically healthy, with the absence of a family history of cardiovascular disease also strongly determining metabolic health.

Relative risk of coronary artery disease in people who have a family history of this disease, estimated based on the literature review, ranged from 0.8 to 2.2 [12]. Similarly, our study showed in each model that the odds of being metabolically healthy were 1.5–2 times greater with no family history of cardiovascular disease. It must be emphasized, however, that family health history reflects not only genetic predisposition to a disease, but also behavioral risk factors. High polygenic risk (top 20% of polygenic risk score) was associated with 1.9-fold odds of developing coronary artery disease [13].

Former smokers were less likely to be metabolically healthy compared to current smokers. Many people who have quit smoking tend to increase calorie intake, which results in weight gain [14] and increased insulin resistance [15]. Some studies have confirmed a higher risk of developing MetS following smoking cessation [16]. The studies carried out among Japanese men demonstrated that smoking cessation reduced the estimated 10-year risk of coronary artery disease; however, weight gain temporarily diminished the beneficial effects of cessation [17]. Thus, if individuals comprising our study population had quit smoking relatively recently and/or gained weight, the negative effect of smoking cessation might still have been visible.

The results of studies on the relationships between alcohol consumption and the prevalence of MetS and its components are ambiguous and may vary depending on sex and interactions with other health behaviors [18]. An increased risk of MetS, resulting from heavy drinking, stems from the influence of large amounts of alcohol on, in particular, blood pressure [19], blood sugar [20], and TG concentrations [21]. In women, only the relationship between alcohol consumption and MetS in the unadjusted model proved significant. Some behavioral risk factors, such as smoking and drinking alcohol, are less pronounced and diverse in women than in men [22]. Our study confirmed that the proportion of women drinking large amounts of alcohol was relatively low [only 2.2% of women declared drinking >15 g ethanol per day,

**Table 4. Factors associated with metabolic health.** Multivariate analyses with backward stepwise selection results for ≥2 components of MetS in men and women.

| Lifestyle factor | | Men | | Women | |
|---|---|---|---|---|---|
| | | OR (95%CI) | p | OR (95%CI) | p |
| family history of cardiovascular disease | yes | **1.0** | - | **1.0** | - |
| | no | **1.87 (1.10–3.17)** | **0.021** | **1.58 (1.27–1.96)** | **<0.001** |
| smoking | current smoker | 1.0 | - | 1.0 | - |
| | former smoker | 0.63 (0.42–0.95) | 0.029 | **0.78 (0.61–0.99)** | **0.038** |
| | never-smoker | 1.10 (0.74–1.63) | 0.634 | 0.93 (0.75–1.15) | 0.482 |
| consumption of alcohol | >30 g | 1.0 | - | 1.0 | - |
| | ≤30 g | **3.28 (1.52–7.10)** | **0.002** | - | - |
| | 0 g | **4.52 (1.87–10.91)** | **0.001** | - | - |
| consumption of coffee | none | 1.0 | - | 1.0 | - |
| | ≤2 servings/ day | - | - | 1.21 (0.92–1.59) | 0.169 |
| | >2 servings/ day | - | - | 1.84 (1.38–2.45) | <0.001 |
| consumption of sweets and sugary drinks | Q1 | **0.53 (0.34–0.83)** | **0.006** | 0.92 (0.74–1.16) | 0.492 |
| | Q2 | 0.79 (0.79–1.19) | 0.257 | 0.99 (0.79–1.25) | 0.945 |
| | Q3 | **0.65 (0.43–0.99)** | **0.043** | 0.80 (0.63–1.01) | 0.063 |
| | Q4 | 1.0 | - | 1.0 | - |
| consumption of animal fats | >1 time/day | - | - | 1.0 | - |
| | ≤1 time/day | - | - | **1.24 (1.05–1.47)** | **0.011** |
| consumption of fruit and vegetables | Q1 | 1.0 | - | 1.0 | - |
| | Q2 | **0.55 (0.35–0.85)** | **0.008** | 0.90 (0.72–1.12) | 0.342 |
| | Q3 | 0.81 (0.55–1.21) | 0.307 | **0.72 (0.57–0.91)** | **0.007** |
| | Q4 | **0.64 (0.42–0.97)** | **0.035** | 0.87 (0.69–1.09) | 0.215 |
| MVPA | ≤ 30 minutes/day | 1.0 | - | 1.0 | - |
| | >30 minutes to ≤ 1 hour/day | - | - | 1.17 (0.92–1.47) | 0.201 |
| | >1 hour to ≤ 2 hours/ day | - | - | 1.06 (0.83–1.36) | 0.628 |
| | >2 hours/ day | - | - | **1.33 (1.05–1.68)** | **0.020** |
| sleep duration | <7 hours/day | 1.0 | - | 1.0 | - |
| | 7–8 hours/day | | | **1.24 (1.02–1.51)** | **0.034** |
| | <7 hours/day | | | 0.69 (0.47–1.02) | 0.060 |

OR–odds ratio, CI–confidence interval, p—value, Q–quintile, MVPA—moderate-to-vigorous physical activity.

**Table 5. Factors associated with metabolic health.** Multivariate analyses with backward stepwise selection results for ≥1 components of MetS in men and women.

| Lifestyle factor | | Men | | Women | |
|---|---|---|---|---|---|
| | | OR (95%CI) | p | OR (95%CI) | p |
| family history of cardiovascular disease | yes | **1.0** | - | **1.0** | - |
| | no | **1.83 (1.08–3.09)** | **0.024** | **1.47 (1.19–1.82)** | **<0.001** |
| smoking | current smoker | 1.0 | - | 1.0 | - |
| | former smoker | 0.68 (0.46–1.02) | 0.060 | **0.77 (0.61–0.98)** | **0.030** |
| | never-smoker | 1.09 (0.75–1.60) | 0.656 | 0.91 (0.74–1.12) | 0.394 |
| consumption of alcohol | >30 g | 1.0 | - | 1.0 | - |
| | ≤30 g | **3.93 (1.64–9.42)** | **0.002** | - | - |
| | 0 g | **3.04 (1.41–6.56)** | **0.005** | - | - |
| consumption of coffee | none | 1.0 | - | 1.0 | - |
| | ≤2 servings/ day | - | - | 1.21 (0.92–1.58) | 0.170 |
| | >2 servings/ day | - | - | **1.72 (1.30–2.28)** | **<0.001** |
| consumption of sweets and sugary drinks | Q1 | **0.55 (0.35–0.86)** | **0.008** | - | - |
| | Q2 | 0.77 (0.52–1.15) | 0.199 | - | - |
| | Q3 | **0.66 (0.44–0.99)** | **0.045** | - | - |
| | Q4 | 1.0 | - | 1.0 | - |
| consumption of animal fats | >1 time/day | 1.0 | - | 1.0 | - |
| | ≤1 time/day | - | - | 1.17 (1.00–0.38) | 0.056 |
| consumption of fruit and vegetables | Q1 | 1.0 | - | 1.0 | - |
| | Q2 | **0.58 (0.38–0.90)** | **0.015** | 0.87 (0.69–1.08) | 0.201 |
| | Q3 | 0.86 (0.58–1.27) | 0.453 | **0.69 (0.55–0.87)** | **0.002** |
| | Q4 | 0.67 (0.44–1.02) | 0.061 | 0.84 (0.67–1.05) | 0.125 |
| MVPA | ≤ 30 minutes/day | 1.0 | - | 1.0 | - |
| | >30 minutes to ≤ 1 hour/ day | - | - | 1.15 (0.92–1.45) | 0.226 |
| | >1 hour to ≤ 2 hours/day | - | - | 1.03 (0.81–1.31) | 0.812 |
| | >2 hours/ day | - | - | **1.26 (1.01–1.59)** | **0.049** |
| sleep duration | <7 hours/day | 1.0 | - | 1.0 | - |
| | 7–8 hours/day | - | - | **1.23 (1.01–1.49)** | **0.037** |
| | <7 hours/day | - | - | 0.72 (0.49–1.05) | 0.089 |

OR–odds ratio, CI–confidence interval, p—value, Q–quintile, MVPA—moderate-to-vigorous physical activity.

while as many as 10.0% of men declared drinking >30 g ethanol per day]. Therefore, the influence of other factors, including dietary, could have been more significant in women.

Numerous studies showed an inverse relationship between coffee consumption and prevalence of MetS and its components [23, 24]. In our analysis, conducted for both sexes separately, the relationship appeared to be significant only in women, which is consistent with the results published by other authors [22, 25]. Although in the previous study carried out in Poland, the relationships between coffee consumption and prevalence of MetS and elevated blood pressure were found in both sexes, similar relationships between coffee consumption and TG and HDL concentrations were demonstrated only in women [26]. Differences in HDL levels following coffee consumption between sexes may result from physiological differences. Some studies suggest that consumption of coffee and other drinks containing caffeine has a positive effect on plasma concentrations of female sex hormones in women, which may lead to an increase in HDL concentrations [27].

Contrary to expectations, our study found relationships between the consumption of certain product groups (fruit and vegetables, sweets and sugary drinks) and metabolic health. The use of a plant-based diet is widely recommended to reduce the risk of cardiometabolic diseases. However, it is well-known that not all diets that focus primarily on plant products are equally beneficial for health [28]. Those containing large amounts of low-quality plant foods, such as refined grains, potatoes (including fries), high-salt foods, and foods and drinks with added sugars, were associated with increased risk of lipid disorders, type 2 diabetes, and coronary artery disease [29, 30]. Previous analyses have confirmed the presence of a "traditional-carbohydrate" dietary pattern in our population, which is characterized by, among others, high consumption of potatoes and refined grains and was linked to high prevalence of abdominal obesity and triglyceridemia [20]. The unexpected inverse relationship between the consumption of fruits and vegetables and the absence of MetS components found in our studies may also arise from the difficulty in determining the causal relationship in cross-sectional studies. Higher intake of these products in individuals with components of MetS may also have resulted from changes in diet, e.g, requested by a doctor due to the prior diagnosis of metabolic disorders. Healthy people and those without a family history of cardiovascular disease could have assumed that their diet was accurate, even though it was not in accordance with dietary guidelines. A similar pattern may also apply to higher consumption of sweets by healthy people. In addition, healthy subjects were found to spend considerably more time on MVPA. Their higher energy requirements could have been fulfilled, at least partially, by consuming more products containing sugar. Similar to our study, a prospective study showed no significant relationships between the consumption of whole grains and health outcomes in 21 countries [31]. High intake of red meat is a widely accepted risk factor of cardiovascular disease and mortality. Some studies, however, have not confirmed such a relationship with respect to MetS and its components [32], which is in accordance with our results. A meta-analysis of prospective studies showed a significant association between unprocessed red meat and mortality risk in the USA, but not in European and Asian populations [33]. One study even found that consumption of ≥3 servings of red meat did not affect lipid and lipoprotein concentrations, nor blood pressure, and resulted in higher HDL levels [34]. These inconclusive results may be due to the cross-sectional character of some studies but may also be the result of the opposing effects of red meat consumption on individual components of MetS, i.e., its beneficial effect on HDL [34] and sometimes on triglyceride levels [35]. Women who limited their consumption of animal fats showed a lower risk of metabolic disorders [36]. However, recent studies on this subject have produced inconsistent results. Although the replacement of saturated fatty acids (SFA) by mono- and polyunsaturated fatty acids improved HDL to low-density lipoprotein (LDL) ratio, reduced total fat intake and SFA not always reduced the risk of cardiovascular

disease [37]. Furthermore, lower mortality risk was observed when SFA, trans fats, or refined carbohydrates were replaced by monounsaturated fatty acids from plants, but not animal products [38].

Our study showed that odds of being healthy were associated with MVPA, which appeared to be consistent with other research [39]. The inverse relationship between sitting time and metabolic health was found only in men. Women tended to have less components of MetS with longer sitting time. This may stem from the influence of education, which modified other risk factors. Numerous studies have demonstrated that greater awareness and higher standards of living, associated with higher education level, generally lead to better health [40], especially among women [41]. In addition, analyses confirmed that in the study group, mostly women with the highest education (45.21%), who worked in a sitting position, reported the longest sitting time (>6 hours/day). In groups reporting sitting time from 3 to ≤6 and ≤3 hours/day, the proportion of women with higher education accounted for 28.41% and 24.93%, respectively.

Women who reported sleeping for 7–8 hours had significantly higher odds of not having metabolic disorders, whereas those sleeping for ≥9 hours had the lowest. Our previous studies showed that longer sleep duration (≥9 hours) was mainly associated with abdominal obesity [42]. Smiley et al. [43] found a U-shaped relationship between the severity of MetS and duration of sleep in women and a semi-linear association in men. There is a risk that individuals who sleep longer have a lower energy expenditure due to staying longer in a lying position. They therefore spend less time doing physical activity, which increases the risk of obesity and MetS [44, 45]. Moreover, longer sleeping time is often associated with sleep fragmentation, which may have a negative effect on metabolic health. Some studies also suggest that sleep disorders are more frequent in women than in men because of psychological factors [46]. This may result in different metabolic effects in both sexes caused by inadequate sleep duration.

## Limitations

The main limitation of this study is its cross-sectional character, which does not allow identification of causal relationships between the analyzed variables. Prior diagnosis of diseases or metabolic disorders could have contributed to significant changes in some participants' lifestyles. Another limitation is the assessment of lifestyle by self-report. However, a validated tools were used in this study. Another limitation was that the FFQ used in the study did not allow for a precise calculation of the calorie and nutrient content of the participants' daily rations.

## Strengths

The key strengths of this study are the large number of participants, homogeneity of the study population regarding age and ethnicity, consideration of several different elements of lifestyle, and analyses conducted in three models, i.e., comparing healthy individuals to those with ≥3, ≥2, and ≥1 components of MetS according to IDF criteria [4].

## Conclusions

The absence of a family history of cardiovascular disease is a very strong determinant of metabolic health. Beneficial gene variants, which promote metabolic health, may probably balance some diet mistakes. However, the results of our study clearly show an association between a healthy lifestyle and being metabolically healthy, defined as the absence of any component of MetS. In both sexes, MVPA performed for >2 hours/day increased the odds of not having metabolic disorders. This was also observed for duration of sleep of 7–8 hours a day, reduction

in animal fat consumption to <2 servings/day, and drinking >2 cups of coffee per day in women as well as abstinence from alcohol, reduction in alcohol consumption to <30 ethanol/ day, and in sitting time to <6 hours/day in men. The odds of being metabolically healthy were, however, significantly lower in former smokers. Some lifestyle modifications may appear effective not only in the prevention of MetS (≥3 components), but also in the case of single metabolic risk factors.

Many studies indicate that East-Central Europe is a region with a very high burden of CVD, and that there are still major disproportions between the countries of Eastern and Western Europe [47–49]. Cardiovascular disease is not only a clinical problem, but also a considerable social and financial burden for these countries. Reducing this high risk of CVD requires not only improving the quality and accessibility of health care and an appropriate medical policy on the part of the state, but also designing and implementing effective prophylactic programs. Actions undertaken both in the past and today have not been sufficient to counteract the high risk of CVD in these populations. The approach used in our study, which focuses on lifestyle elements that help to maintain metabolic health (i.e. preventing risk factors by modifying one's lifestyle before they appear) follows the concept of ideal cardiovascular health promoted by the American Heart Association [50, 51]. This approach can also be used as the leading strategy in the design of innovative prophylactic and educational programs aimed at reinforcing healthy behavior among society.

What is already known on this subject?

- Current literature emphasizes that there is no single, universally accepted definition of metabolic health.

- Although several variants of different genes are involved in the pathogenesis of metabolic syndrome, all of its components are influenced by lifestyle. It is not clear which elements of lifestyle are the strongest predictors of metabolic health in adults.

- What does this study add?

- Family history of cardiovascular disease is a strong determinant of metabolic health.

- Healthy lifestyle increased the odds of being metabolically healthy, defined as the absence of any component of metabolic syndrome.

- MVPA performed for >2 hours/day increased the odds of not having metabolic disorders.

- When developing prevention programs aiming to maintain metabolic health, physiological differences and possible diversity of lifestyle in both sexes need to be taken into account.

## Supporting information

**S1 File. Healthy basic.**
(PDF)

## Author Contributions

**Conceptualization:** Edyta Suliga, Elzbieta Ciesla, Agnieszka Piechowska, Stanislaw Gluszek.

**Formal analysis:** Edyta Suliga, Elzbieta Ciesla, Magdalena Lelonek.

**Funding acquisition:** Edyta Suliga, Stanislaw Gluszek.

**Methodology:** Elzbieta Ciesla, Stanislaw Gluszek.

**Supervision:** Edyta Suliga, Stanislaw Gluszek.

**Writing – original draft:** Edyta Suliga, Elzbieta Ciesla, Magdalena Lelonek, Agnieszka
   Piechowska.

**Writing – review & editing:** Edyta Suliga, Magdalena Lelonek.

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
