## [Decision Letter · Decision Letter 0]

22 Jun 2022

PONE-D-22-07560Lifestyle elements associated with metabolic health in adultsPLOS ONE

Dear Dr. Lelonek,

Thank you for submitting your manuscript to PLOS ONE. After careful consideration, we feel that it has merit but does not fully meet PLOS ONE’s publication criteria as it currently stands. Therefore, we invite you to submit a revised version of the manuscript that addresses the points raised during the review process.

We look forward to receiving your revised manuscript.

Kind regards,

Hsin-Yen Yen

Academic Editor

PLOS ONE

Journal Requirements:

“We thank the Maria Sklodowska-Curie Institute of Oncology in Warsaw (Poland) and the Polish-Norwegian Foundation Research Fund for suport.”

“This work was supported under the program of the Minister of Science and Higher Education under the name “Regional Initiative of Excellence” in 2019–2022, project number: 024/RID/2018/19, financing amount: 11.999.000,00 PLN.”

“This work was supported under the program of the Minister of Science and Higher Education under the name “Regional Initiative of Excellence” in 2019–2022, project number: 024/RID/2018/19, financing amount: 11.999.000,00 PLN.”

Reviewers' comments:

Reviewer's Responses to Questions

**Comments to the Author**

1. Is the manuscript technically sound, and do the data support the conclusions?

Reviewer #1: Partly

Reviewer #2: Partly

2. Has the statistical analysis been performed appropriately and rigorously? 

Reviewer #1: Yes

Reviewer #2: I Don't Know

3. Have the authors made all data underlying the findings in their manuscript fully available?

Reviewer #1: Yes

Reviewer #2: Yes

4. Is the manuscript presented in an intelligible fashion and written in standard English?

Reviewer #1: Yes

Reviewer #2: Yes

5. Review Comments to the Author

Reviewer #1: Typo at bottom of page 3, 'Due differences' probably should be 'Due to differences'

Need more information in the methods about how the specifics of coffee and food was collected. Was the coffee black or did it include creamers? Response would be very different and potentially contribute to sugar contact.

I have concerns with the language related to family history of cardiovascular disease throughout. Poor lifestyle often contributes to an older family member being diagnosed with disease. This needs to be better clarified and the claims about CV disease being a strong determinant of metabolic health needs to be fixed to be stated as family history being a risk factor and not a determinant.

Tables 3, 4 , and 5 are too big with too much data that isn't necessary.

Missing a reference in the Discussion section, page 15, third paragraph about those who quit smoking tend to increase calorie intake - include citation for this. Also in the Discussion, paragraph 4 states that this study confirms.... but it can't because it all associations and looking at risk for metabolic syndrome. Fix this claim.

Page 16 needs a sentence re-worded - 'women who limited animal fats were more likely not to have metabolic disorders.' This is difficult to read, re-word to a positive flow.

Page 16 needs a reference for the statement that people who sleep longer, in a lying position longer, have a higher risk. This seems like a stretch, were is the reference?

I also recommend changing the title to include the word 'risk' somewhere as the methods and data are all risk of development.

Would be good to have more connection to why this work is needed and how it adds to the literature as it is already well-known that healthy behaviors can prevent and treat diseases.

Reviewer #2: Interesting research about several health components associated with metabolic health (absence of MetS components) in adults aged 40-65 years. This approach is interesting, considering that the majority of the articles associated lifestyle factors to MetS components.

Major:

- All the statistical models are unadjusted? Potential confounders, and interactions between the different components should be addressed. In the case that the models are adjusted, the adjustment variables and the description of the variables used should be included.

- Dietary assessment should be described. How nutritional data was collected? Using FFQ? 24R? 3-days register? Please, provide this information in the methods section.

- Baseline characteristics of the study participants should be included, including also medication status.

- Is it possible to include nutritional data (caloric intake, fiber, fatty acids profile)? Total energy intake should be included as a covariate in the adjustment.

Minor:

- Tables are not easily readable. Maybe it is necessary to add borders in order to clarify the data (each factor, quartiles, etc.).

- Table 3: Use the same order to present data (Q4, Q3, Q2, Q1 or Q1 to Q4).

- Limitations: only physical activity was self-reported? Please add if dietary data, smoking status and other health outcomes were self-reported.

6. PLOS authors have the option to publish the peer review history of their article (what does this mean?). If published, this will include your full peer review and any attached files.

Reviewer #1: No

Reviewer #2: No

---

## [Author Response · Author response to Decision Letter 0]

12 Jul 2022

Response to Reviewers

Comments Reviewer 1

Section: Anthropometric measurements, assessment of biomarkers, blood pressure

Comments and Suggestions for Author(s): Typo at bottom of page 3, 'Due differences' probably should be 'Due to differences'.

Our answer: The typo has been corrected.

Section: Anthropometric measurements, assessment of biomarkers, blood pressure and lifestyle data

Comments and Suggestions for Author(s): Need more information in the methods about how the specifics of coffee and food was collected. Was the coffee black or did it include creamers? Response would be very different and potentially contribute to sugar contact. 

Our answer: The respondents were only asked about the amount and frequency of coffee consumption in general. The consumption of sugar and other products was assessed in separate questions.

Section: Family history of cardiovascular disease

Comments and Suggestions for Author(s): I have concerns with the language related to family history of cardiovascular disease throughout. Poor lifestyle often contributes to an older family member being diagnosed with disease. This needs to be better clarified and the claims about CV disease being a strong determinant of metabolic health needs to be fixed to be stated as family history being a risk factor and not a determinant. 

Our answer: Information concerning the family history of CVD has been corrected.

Section: Results

Comments and Suggestions for Author(s): Tables 3, 4 , and 5 are too big with too much data that isn't necessary. 

Our answer: We agree that these tables are too big. Their size results from the fact that they contain the OR with 95% CI obtained across groups with different numbers of components of MetS. The aim of the backward stepwise analysis used in our study was to present the final model, which consisted of the best possible set of predictors affecting the participants’ metabolic health, without losing the significance of the model. We decided to include all analyzed predictors, even when the ORs were statistically insignificant. In our opinion, this presentation of data is clearer to the readers. However, if the Reviewer believes that the tables should be revised, we can, for example, delete the 95% CI or the variables for which p >0.05 in order to reduce the amount of data shown in the tables; instead, we can provide these data somewhere else, for instance in the table legends.

Section: Discussion

Comments and Suggestions for Author(s): Missing a reference in the Discussion section, page 15, third paragraph about those who quit smoking tend to increase calorie intake - include citation for this. Also in the Discussion, paragraph 4 states that this study confirms.... but it can't because it all associations and looking at risk for metabolic syndrome. Fix this claim.

Our answer: As per the Reviewers suggestion, the missing references were added.

Section: Discussion

Comments and Suggestions for Author(s): Page 16 needs a sentence re-worded - 'women who limited animal fats were more likely not to have metabolic disorders.' This is difficult to read, re-word to a positive flow. 

Our answer: The sentence has been corrected.

Section: Discussion

Comments and Suggestions for Author(s): Page 16 needs a reference for the statement that people who sleep longer, in a lying position longer, have a higher risk. This seems like a stretch, were is the reference?

Our answer: As per the Reviewers suggestion, the missing references have been added.

Section: Title

Comments and Suggestions for Author(s): I also recommend changing the title to include the word 'risk' somewhere as the methods and data are all risk of development.

Our answer: We decided not to include the word ‘risk’ in the title, because the primary aim of our study was to investigate which elements of lifestyle are associated with metabolic health (absence of MetS components) in adults aged 40-65 years. Therefore, our aim was to find health determinants, rather than risk factors.

Our approach, which focuses on elements of lifestyle that help to maintain metabolic health, follows the concept of ideal cardiovascular health promoted by the American Heart Association [Lloyd-Jones et al. 2010; Labarthe2012; Tsao et al. 2022]. This approach can also be used as the leading strategy in the design of innovative prophylactic and educational programs aimed at reinforcing healthy behavior among society.

Section: Conclusions

Comments and Suggestions for Author(s): Would be good to have more connection to why this work is needed and how it adds to the literature as it is already well-known that healthy behaviors can prevent and treat diseases.

Our answer: As per the Reviewer’s suggestion, we have added more information to the summary about why the work is needed and its contribution to the literature.

Comments Reviewer 2

Section: Statistical methods/ Results

Comments and Suggestions for Author(s): All the statistical models are unadjusted? Potential confounders, and interactions between the different components should be addressed. In the case that the models are adjusted, the adjustment variables and the description of the variables used should be included.

Our answer: Univariate analyses showed the effect of individual factors on a group of participants with 0, 1, 2, 3 or more MetS components, and were not adjusted. Conversely, the aim of the backward stepwise analysis was to present the final model, which consisted of the best possible set of predictors that affected the participants’ metabolic health, without losing the significance of the model. The vast number of variables and cases allowed us to use the aforementioned procedure. This is why all potential variables that may have constituted the group of covariates were shown in both types of regression models.

Section: Anthropometric measurements, assessment of biomarkers, blood pressure and lifestyle data

Comments and Suggestions for Author(s): Dietary assessment should be described. How nutritional data was collected? Using FFQ? 24R? 3-days register? Please, provide this information in the methods section.

Our answer: Description of the dietary assessment has been added, as per the Reviewer’s suggestion. The description is also provided below. Information on smoking, consumption of alcohol, coffee and other food products, physical activity, sedentary behavior, and sleep duration was collected during face-to-face interviews. Dietary data were collected using the Frequency Questionnaire (FFQ), which consisted of a list of 67 products. The questionnaire was constructed based on a previously designed and validated FFQ for the Polish arm of the PURE study. The questionnaire was highly accurate and repeatable compared to the reference method [Dehghan, M.; Ilow, R.; Zatonska, K.; Szuba, A.; Zhang, X.; Mente, A.; Regulska-Ilow, B. Development, reproducibility and validity of the food frequency questionnaire in the Poland arm of the Prospective Urban and Rural Epidemiological (PURE) study. J. Hum. Nutr. Diet. 2012, 25, 225–232.]. The respondents declared how frequently they consumed a standard portion of each product over the preceding year.

Section: Study population/ The definition of metabolic health/ Results

Comments and Suggestions for Author(s): Baseline characteristics of the study participants should be included, including also medication status

Our answer: The Study section describes the participants’ baseline characteristics, including medication status. This information was also taken into account when the patients were qualified for the presence of each MetS component, in accordance with the IDF definition of MetS.

Section: Anthropometric measurements, assessment of biomarkers, blood pressure and lifestyle data/

Limitations

Comments and Suggestions for Author(s): Is it possible to include nutritional data (caloric intake, fiber, fatty acids profile)? Total energy intake should be included as a covariate in the adjustment. 

Our answer: The questionnaire we used in our study did not allow for a precise calculation of the calorie content or the content of other nutrients in the participants’ diet. This information has been provided in the limitations of the study.

Section: Results

Comments and Suggestions for Author(s): Tables are not easily readable. Maybe it is necessary to add borders in order to clarify the data (each factor, quartiles, etc.).

Our answer: As per the Reviewer’s suggestion, for better readability, borders between the rows and columns with data have been added to Tables 1 and 2.

Section: Results

Comments and Suggestions for Author(s): Table 3: Use the same order to present data (Q4, Q3, Q2, Q1 or Q1 to Q4).

Our answer: The order of the data has been changed according to the Reviewer’s suggestion.

Section: Anthropometric measurements, assessment of biomarkers, blood pressure and lifestyle data

Comments and Suggestions for Author(s): Limitations: only physical activity was self-reported? Please add if dietary data, smoking status and other health outcomes were self-reported.

Our answer: All lifestyle data were self-reported. The relevant information has been added, as per the Reviewer’s suggestion.

---

## [Decision Letter · Decision Letter 1]

26 Jul 2022

PONE-D-22-07560R1Lifestyle elements associated with metabolic health in adultsPLOS ONE

Dear Dr. Lelonek,

Thank you for submitting your manuscript to PLOS ONE. After careful consideration, we feel that it has merit but does not fully meet PLOS ONE’s publication criteria as it currently stands. Therefore, we invite you to submit a revised version of the manuscript that addresses the points raised during the review process.

We look forward to receiving your revised manuscript.

Kind regards,

Hsin-Yen Yen

Academic Editor

PLOS ONE

Journal Requirements:

Reviewers' comments:

Reviewer's Responses to Questions

**Comments to the Author**

1. If the authors have adequately addressed your comments raised in a previous round of review and you feel that this manuscript is now acceptable for publication, you may indicate that here to bypass the “Comments to the Author” section, enter your conflict of interest statement in the “Confidential to Editor” section, and submit your "Accept" recommendation.

Reviewer #1: (No Response)

Reviewer #2: All comments have been addressed

2. Is the manuscript technically sound, and do the data support the conclusions?

Reviewer #1: Yes

Reviewer #2: No

3. Has the statistical analysis been performed appropriately and rigorously? 

Reviewer #1: Yes

Reviewer #2: I Don't Know

4. Have the authors made all data underlying the findings in their manuscript fully available?

Reviewer #1: Yes

Reviewer #2: Yes

5. Is the manuscript presented in an intelligible fashion and written in standard English?

Reviewer #1: Yes

Reviewer #2: Yes

6. Review Comments to the Author

Reviewer #1: Most reviewer comments were addressed. There are few that I recommend be further considered before publication.

More inclusion of risk needs to be considered. The purpose or aim is stated to be about association but then the conclusion is about risk of developing metabolic syndrome. Consider adjusting the title or at least adding the work risk to the key words list.

Title suggestion, Lifestyle elements and risk of metabolic syndrome in adults

Discussion

Consider the following suggestion for this section

Discussion The results of our study confirmed that a healthy lifestyle may significantly increase the odds of being metabolically healthy, with the absence of family history of cardiovascular disease also strongly determining metabolic health.

Suggestion, instead of study confirmed consider changing to study add support that

Conclusion

Consider the following suggestion for this section

However, the results of our study clearly show that a healthy lifestyle increased the odds of being metabolically healthy, defined as the absence of any component of MetS.

Suggestion, if your study aim was to find association, don’t conclude that your study clearly show that a healthy lifestyle increased the odds…should be clearly show an association.

Reviewer #2: Thank you for your answer. I'm not familiar with the statistical approach that you are using. However, i don't think that these 5 long tables help us to understand the association between life-style factors and MetS severity. Two of them are only frequencies, not relationships. In the case of table 3, the strongest association is observed in alcohol intake (OR: 5.49 (2.23-13.52) <0.001). So, in this OR and 95% CI, you are also considering the other significant factors such as family history of CVD, smoking status, or sweet beverages intake?? Probably, multiple comparison correction should be applied.

7. PLOS authors have the option to publish the peer review history of their article (what does this mean?). If published, this will include your full peer review and any attached files.

Reviewer #1: No

Reviewer #2: No

---

## [Author Response · Author response to Decision Letter 1]

6 Sep 2022

Response to the Reviewers

Reviewer #1: 

Comments and Suggestions for Author(s): Most reviewer comments were addressed. There are few that I recommend be further considered before publication.

More inclusion of risk needs to be considered. The purpose or aim isstated to be about association but then the conclusion is about risk of developing metabolic syndrome. Consider adjusting the title or at least adding the work risk to the key words list.

Title suggestion, Lifestyle elements and risk of metabolic syndrome in adults

Our answer:

We have changed the title. Now the title is ‘Lifestyle elements and risk of metabolic syndrome in adults’. We have also added ‘risk factors’ to the list of key words.

Comments and Suggestions for Author’s (Discussion): Consider the following suggestion for this section

The results of our study confirmed that a healthy lifestyle may significantly increase the odds of being metabolically healthy, with the absence of family history of cardiovascular disease also strongly determining metabolic health.

Suggestion, instead of study confirmed consider changing to study add support that

Our answer:

We have tailored the Discussion section to the Reviewer’s suggestion.

The sentence currently reads: The results of our study show that a healthy lifestyle may significantly increase the odds of being metabolically healthy, with the absence of a family history of cardiovascular disease also strongly determining metabolic health.

Comments and Suggestions for Author’s (Conclusion)

Consider the following suggestion for this section:

However, the results of our study clearly show that a healthy lifestyle increased the odds of being metabolically healthy, defined as the absence of any component of MetS. Suggestion, if your study aim was to find association, don't conclude

that your study clearly show that a healthy lifestyle increased the odds…should be clearly show an association.

Our answer:

The sentence has been corrected. It now reads: However, the results of our study clearly show an association between a healthy lifestyle and being metabolically healthy, defined as the absence of any component of MetS.

Reviewer #2:

Comments and Suggestions for Author’s: 

Thank you for your answer. I'm not familiar with the statistical approach that you are using. However, i don't think that

these 5 long tables help us to understand the association between life- style factors and MetS severity. Two of them are only frequencies, not relationships. In the case of table 3, the strongest association is observed in alcohol intake (OR: 5.49 (2.23-13.52) <0.001). So, in this OR and 95% CI, you are also considering the other significant factors such as family history of CVD, smoking status, or sweet beveragesintake?? Probably, multiple comparison correction should be applied. 

Our answer:

According to the reviewer’s suggestions, we showed only multivariate analyses with backward stepwise selection. Univariate analyses were performed, but the data were not shown in order to make the tables more readable and much shorter.

---

## [Decision Letter · Decision Letter 2]

20 Sep 2022

Lifestyle elements and risk of metabolic syndrome in adults

PONE-D-22-07560R2

Dear Dr. Lelonek,

We’re pleased to inform you that your manuscript has been judged scientifically suitable for publication and will be formally accepted for publication once it meets all outstanding technical requirements.

Kind regards,

Hsin-Yen Yen

Academic Editor

PLOS ONE

Additional Editor Comments (optional):

Reviewers' comments:

Reviewer's Responses to Questions

**Comments to the Author**

1. If the authors have adequately addressed your comments raised in a previous round of review and you feel that this manuscript is now acceptable for publication, you may indicate that here to bypass the “Comments to the Author” section, enter your conflict of interest statement in the “Confidential to Editor” section, and submit your "Accept" recommendation.

Reviewer #1: All comments have been addressed

2. Is the manuscript technically sound, and do the data support the conclusions?

Reviewer #1: (No Response)

3. Has the statistical analysis been performed appropriately and rigorously? 

Reviewer #1: (No Response)

4. Have the authors made all data underlying the findings in their manuscript fully available?

Reviewer #1: (No Response)

5. Is the manuscript presented in an intelligible fashion and written in standard English?

Reviewer #1: (No Response)

6. Review Comments to the Author

Reviewer #1: (No Response)

7. PLOS authors have the option to publish the peer review history of their article (what does this mean?). If published, this will include your full peer review and any attached files.

Reviewer #1: No

---

## [Editor Report · Acceptance letter]

22 Sep 2022

PONE-D-22-07560R2 

 Lifestyle elements and risk of metabolic syndrome in adults 

Dear Dr. Lelonek:

I'm pleased to inform you that your manuscript has been deemed suitable for publication in PLOS ONE. Congratulations! Your manuscript is now with our production department. 

Kind regards, 

on behalf of

Dr. Hsin-Yen Yen 

Academic Editor

PLOS ONE